# Efficiency of Nitrogen Use in Sunflower

**DOI:** 10.3390/plants11182390

**Published:** 2022-09-14

**Authors:** Ester dos Santos Coêlho, Almir Rogério Evangelista de Souza, Hamurábi Anizio Lins, Manoel Galdino dos Santos, Matheus de Freitas Souza, Francilene de Lima Tartaglia, Anna Kézia Soares de Oliveira, Welder de Araújo Rangel Lopes, Lindomar Maria Silveira, Vander Mendonça, Aurélio Paes Barros Júnior

**Affiliations:** 1Department of Agronomic and Forest Sciences, Universidade Federal Rural do Semi-Árido, Mossoró 59625900, RN, Brazil; 2Departament of Agronomic Engineering, Instituto Federal de Alagoas, Piranhas 57460000, AL, Brazil; 3Department of Agronomy, Universidade de Rio Verde, Rio Verde 75901970, GO, Brazil

**Keywords:** *Helianthus annuus* L., nitrogen fertilization, phenotypic plasticity

## Abstract

The large variation in the response of sunflower to nitrogen fertilization indicates the need for studies to better adjust the optimum levels of this nutrient for production conditions. Our objectives were to analyze the agronomic yield of sunflower cultivars as a function of nitrogen fertilization; indicate the cultivar with high nitrogen use efficiency; and measure the adequate N dose for sunflower through nutritional efficiency. The completely randomized block design with split plots was used to conduct the experiments. The treatments included five nitrogen rates being allocated in the plots and the four sunflower cultivars. To estimate the nutrient use efficiency in the sunflower, we measured agronomic efficiency (AE), physiological efficiency (PE), agrophysiological efficiency (APE), apparent recovery efficiency (ARE), and utilization efficiency (UE). The results indicate that all cultivars had a reduction in AE due to the increase in N doses in the first crop. For PE, the highest values were observed for Altis 99 during the 2016 harvest. In that same harvest, Altis 99 had the highest APE. The dose of 30 kg ha^−1^ provided greater ARE for all cultivars in both crops, with greater emphasis on BRS 122 and Altis 99. The cultivation of cultivars Altis 99 and Multissol at a dose of 30 kg ha^−1^ in is recommended semiarid regions.

## 1. Introduction

Since the domestication of crops, creating agricultural systems with high productivity has been an essential factor for humans [1]. Early studies showed that plant fertilization is a key factor in achieving higher production [2]. These studies culminated in the so-called green revolution, where a large amount of technology and practices were introduced into agricultural systems, resulting in high yields [3,4]. Among the agricultural practices adopted, intense mineral fertilization was an indisputable factor in obtaining high yields [5]. However, the increase in fertilizer costs and the evolution of research work have shown that fertilization should focus on the efficiency of nutrient use for each crop [2,6].

Among the crops that have already shown increased productivity due to the application of mineral nutrients, sunflower (*Helianthus annuus* L.) is an example [7]. This crop is an oilseed with importance in human nutrition and in the energy industry [8]. Sunflower seeds provide about 10% of all edible vegetable oil in the world [8,9,10,11].

In addition, this oil is the main component in the production of biodiesel, reducing the impacts caused by non-renewable energy sources [12,13]. Sunflower has broad phenotypic plasticity for its cultivars, allowing its cultivation in different climate and soil conditions [14,15]. The factor that will determine the productive success of this culture is the adjustment of cultural practices to each region. Thus, the nutritional management of sunflower must be understood among the available cultivars and their adaptability to each in producing region [10,16,17,18].

Nitrogen (N) is one of the essential nutrients for sunflower growth and development. This nutrient constitutes structural and metabolic elements of plant cells, such as amino acids, proteins, nucleic acids, and enzymes [2,9,19,20,21,22]. Sunflower absorbs nitrogen mainly in inorganic forms such as ammonium (NH_4_^+^) and nitrate (NO_3_^−^) [10]. After N absorption, there is a stimulus for the vegetative growth of sunflower plants, favoring the synthesis of photoassimilates and by-products for fruit and seed formation [23,24]. Studies have found that seed and oil productivity increased by up to 40% due to supplementation with nitrogen sources [25,26,27].

The application of N below or above the optimum range reduces the productivity or efficiency with which this nutrient is used by the crop [28]. Furthermore, indirect effects on plant development can affect the productivity of harvest. For example, the application of high doses on sunflower stimulated the plants to grow in terms of height, increasing the risk of lodging in areas with excessive winds [24]. Graham and Varco [29] observed that due to the rooting of sunflower plants, the probability of excessive N fertilization is high in the soil. Schatz et al. [30] suggested mean values of 33 kg ha^−1^ of nitrate residue in the soil under an application rate of 90 kg N ha^−1^, also confirming that the excessive application of this nutrient in the soil can cause negative effects on sunflower. The excessive addition of N causes an imbalance between the vegetative and reproductive phases of the crop, stimulating uncontrolled vegetative growth and delaying plant maturation [23]. The combination of these factors reduces sunflower yield and increases the likelihood of pest attack [31,32].

An alternative way to avoid problems related to fertilization outside the optimum range is to determine the efficiency of the use of the nutrient. This efficiency is usually evaluated through performance indices. These indices correlate the applied doses of the nutrient with the agronomic performance variables of the crop [33]. Some examples are agronomic efficiency (AE), physiological efficiency (PE), agrophysiological efficiency (APE), utilization efficiency (UE), and recovery efficiency (RE). Tests can be conducted in the field or under controlled conditions; however, the first has the advantage of encompassing other environmental factors that can affect the efficiency of nutrient use [34]. These environmental factors can be better understood when trials are conducted for at least two seasons. Fertilizers that are well calibrated for crop efficiency can ensure high yields and rationing in the use of high-cost mineral fertilizers such as N [35].

The wide variation in the responses of sunflower to nitrogen fertilization indicates the need for studies to better adjust the optimum levels of this nutrient for different production conditions [36]. In addition to growing conditions, there are differences between cultivars in terms of nitrogen-use efficiency. This fact could be due to the ability of plants to induce different mechanisms during nutrient acquisition and translocation, among other factors that are not fully known [37,38,39]. Schwerz et al. [40] studied the effect of nitrogen fertilization on sunflower cultivars and found that the cultivar Olisun 3 had a higher yield and leaf area index at doses ranging from 80 to 120 kg of N ha^−1^, also resulting in greater crop efficiency for nutrient use. Silva [41] recommended the planting of cultivar BRS 323 at a dose of 60 kg of N ha^−1^ to obtain greater efficiency in the use of N. Thus, we hypothesized that sunflower cultivars in the Brazilian semiarid region have different morphophysiological mechanisms for the efficiency of nitrogen utilization, reflecting a different absorption capacity of this nutrient. The objectives of this work were (1) to analyze the agronomic yield of sunflower cultivars as a function of nitrogen fertilization; (2) to indicate the cultivar that has the best use of nitrogen; and (3) to measure the adequate N dose for sunflower through nutritional efficiency. To verify the hypothesis, we measured indices related to agronomic, physiological, and morphological aspects.

## 2. Results

Rising fertilizer prices promote greater interest in the efficient use of resources [30]. In addition, studies that allow for estimating the adequate use of fertilizers are of great importance for scientists in the field of soil science [38]. Therefore, it is essential to plan agronomic strategies that improve the use of environmental resources [41].

### 2.1. Agronomic Efficiency (AE)

All cultivars had a reduction in AE due to the increase in N doses in the first crop, with a greater reduction (69.6%) for the cultivar Aguará 06 (Figure 1A). In that same harvest, the lowest AE values were observed for doses of 120 kg ha of N, with values of 5.64, 6.29, 3.77, and 7.32 kg kg^−1^ for Aguará 06, Multissol, Altis 99, and BRS 122, respectively (Table 1).

The cultivar Aguará 06 showed higher AE values compared to the other cultivars at doses of 30 (18.53 kg kg^−1^), 60 (14.30), and 90 (12.72) (Table 1). At the dose equivalent to 120 kg ha of N, the EA was similar to the cultivars (Table 1).

In the second crop, the cultivars Multissol and BRS 122 also showed a reduction in AE with the increase in N doses. However, Aguará 06 increased its AE due to the application of higher N doses, ensuring higher AE compared to the other cultivars at the dose of 120 kg ha of N (Figure 1B, Table 1). For Altis 09, there was no effect of doses on AE, with a mean value of 10.20 kg kg^−1^ (Figure 1B).

### 2.2. Physiological Efficiency (PE)

The highest PE values were observed for Altis 99 during the 2016 harvest. This cultivar reached values of 171.85, 146.25, and 186.17, kg kg^−1^ for the doses of 30, 60, and 90 kg of N ha^−1^ in the first crop (Table 2). However, increasing the dose of 120 kg ha^−1^ reduced the PE of Altis 99, equaling the other cultivars tested (Figure 2A, Table 2).

Cultivars Aguará 06, Multissol, and BRS 122 did not change their PE due to increased doses. Furthermore, the EF values among these cultivars were similar for the doses of 60 to 120 kg ha^−1^ of N. Only Aguará 06 obtained a higher PE for a lower dose (30 kg ha^−1^) in 2016 (Figure 2A, Table 2).

In 2016, Altis 99 did not show high superiority for PE compared to other cultivars (Figure 3B). Altis 99 showed a reduction of 68.78% between 2016 and 2017 for an application of 30 kg of N ha^−1^ (Table 2). Cultivars Altis 99 and BRS 122 also showed lower PE in 2017 than 2016. Only Aguará increased its PE in 2017 when 90 kg of N ha^−1^ was applied compared to 2016 (Table 2).

No behavior pattern was observed for PE due to the increase in N dose (Figure 2B).

Cultivars showed similar PE values for most treatments in 2017; only the application of 60 kg ha^−1^ of N resulted in lower PE for BRS 122 compared to the other cultivars (Table 2).

### 2.3. Agrophysiological Efficiency (APE)

The highest agrophysiological efficiency (APE) in the first crop was obtained by Altis 99, with 83.56 kg kg^−1^ at a dose of 30 kg of N ha^−1^ (Figure 3A, Table 3). However, the increase in N doses reduced the EPA of Altis 99, equaling cultivars BRS 122 and Multissol (Figure 3A, Table 3).

Aguará 06 did not show large variations for APE due to the increase in doses (Figure 3A) in the 2016 harvest. Aguará 06 still obtained higher APE compared to the other cultivars considering the doses of 60, 90, and 120 kg ha^−1^ (Table 3).

APE showed a similar behavior to PE during the 2017 harvest due to increased N doses (Figure 3B). However, the APE was greater or equal for some cultivars and doses in 2017. Altis 99 showed a higher APE in 2017 compared to 2016 for all doses tested (Table 3). The APE of BRS 122 in 2017 was also higher for doses of 30 (59.75), 60 (56.64), and 90 kg ha^−1^ (49.87) compared to 2016 (33.10, 42, 87, 24.77, and 30.06). For Aguará 06, the APE was higher at doses of 30 (79.54) and 60 kg ha^−1^ (91.05), while Multissol showed a higher value only at the dose of 60 kg ha^−1^ (83.29) in 2017 compared to 2016 (Table 3).

### 2.4. Apparent Recovery Efficiency (ARE)

In the first agricultural crop, the highest apparent recovery efficiency (ARE) obtained was in the cultivar BRS 122 at a dose of 30 kg of N ha^−1^ (39.63 kg kg^−1^), with a reduction in doses of 60 (22.68) and 90 (19.79) (Table 4, Figure 4A). However, with the increase to the dose of 120 kg of N ha^−1,^ there was a small increase in the BRS 122 ARE (Table 4, Figure 4A). Similar behavior was observed in the cultivar Altis 99, with a higher value in the dose of 30 (39.63), a decrease in the doses of 60 (10.97) and 90 (8.90), and an increase in the dose of 120 (16.21) (Table 4, Figure 4A).

Cultivars Multissol and Aguará 06 decreased as the doses were increased in the first crop: Multissol with 37.89, 19.26, 14.60, and 14.46 kg kg^−1^ at doses of 30, 60, 90, and 120 kg of N ha^−1^, and Aguará 06 with 32.78, 24.17, 22.87, and 10.98 kg kg^−1^ at doses of 30, 60, 90, and 120 kg of N ha^−1^, respectively (Table 4, Figure 4A).

In the second crop, Aguará 06 gradually increased as N was increased, a similar trend was found in cultivar Altis 99 for this efficiency (Table 4, Figure 4B).

The highest ARE value obtained in 2017 was in the cultivar Multissol at a dose of 30 kg of N ha^−1^ (31.33 kg kg^−1^). For this same cultivar, there was an increase in the dose of 90 (22.71) (Table 4, Figure 4B). Cultivar BRS 122 showed similar behavior, with 20.17 and 14.37 at doses of 30 and 90, respectively (Table 4, Figure 4B).

### 2.5. Utilization Efficiency (UE)

Cultivar Altis 99 obtained in the first crop the highest utilization efficiency value (UE) with 33.26 kg kg^−1^ at the dose of 30 kg of N ha^−1^, with a reduction in the dose of 120 (10.83) (Table 5, Figure 5A). A similar behavior was obtained for Aguará 06, with a reduction of 70.95% in the application of the dose of 120 kg of N ha^−1^ and a higher value obtained in the dose of 30 (29.08) demonstrating that high doses of N reduce the efficiency of utilization of these cultivars (Table 5, Figure 5A).

The UE in 2016 showed a behavior similar to that of the ARE, where the cultivars Multissol and BRS 122 presented the highest values of doses of 30 (24.01 and 25.59) and 90 (10.18 and 18.30), respectively (Table 5, Figure 5A). 

In the second agricultural season, Multissol obtained 13 kg kg^−1^ under the application of 30 kg of N ha^−1^, which is the highest value found among the cultivars (Table 5, Figure 5B). In 2017, all cultivars showed a reduction in UE at all doses, with the exception of Aguará 06 at the dose of 120 kg of N ha^−1^ (12.42 kg kg^−1^) (Table 5, Figure 5B).

## 3. Discussion

### 3.1. Agronomic Efficiency (AE) Was Higher at Lower Doses

In order to understand agronomic efficiency (AE), we must take into account the interactions involving biotic and abiotic factors, the dose of available nutrients, and the plant’s metabolic activities that favor nutrient absorption [42]. Furthermore, nitrogen fertilization directly influences achene yield [40]. In the present study, AE stood out in lower N doses, showing that high N doses can reduce agronomic efficiency [42,43]. In sunflower, the excess of N prolongs the vegetative phase, as well as causing a reduction in the productivity of achenes, reducing the oil content in the seed and increasing the protein content [44,45].

The higher AE obtained by the cultivar Aguará 06 at doses of 30 to 90 kg of N ha^−1^ in relation to the other cultivars may have been due to a genetic effect, such as the phenotypic plasticity attributed to this cultivar, which provides differences in the productivity of achenes between cultivars of the same species [40]. Furthermore, the source of applied N (urea) favors the increase in the productivity of achenes in Aguará 06 [40].

The reduction in AE in the cultivars Multissol and BRS 122 with the increment of higher doses indicates that the excess of N causes a decrease in the productivity of achenes, and with it a reduction in the oil content produced by sunflower [46]. With a lower productivity of achenes at higher doses of N, the AE decreases. The lower AE of Aguará 06 in 2016 compared to 2017 for the dose of 30 kg of N ha^−1^ may be associated with lower rainfall in 2017 during the flowering stage (50 DAS). In this phase, sunflower requires greater absorption of water and nutrients for grain filling [47,48], and the occurrence of rain (in addition to irrigation) stimulated greater grain production in 2016.

### 3.2. The Increase in Physiological Efficiency (PE) Is Associated with the Increase in Biomass

The increase in PE in cultivar Altis 99 occurs due to a greater production of biomass in this cultivar, showing that the direction of N favors shoot growth. Parameters involving biomass production are related to the amount of N available to the plant [45]. Thus, in the application of doses between 30 and 90 kg of N ha^−1^, the vegetative growth of cultivar Altis 99 was promoted, and consequently, its physiological efficiency increased.

The similarity observed between the cultivars Aguará 06, Multissol, and BRS 122 in doses from 60 to 120 kg ha^−1^ of N indicates that the N applied in these cultivars presents a similar utilization. A similar PE between cultivars of the same species indicates that they have a similar ability to acquire N to incorporate it into their metabolic functions, such as in biomass production [49].

The reduction presented by Altis 99 and other cultivars in the 2017 harvest can be explained by the low rainfall observed during the flowering season. Under this condition, the adaptation response of the cultivars demonstrates a difference in the use of nitrogen, changing the allocation of this nutrient and biomass in the plant compartments due to a limitation in rainfall or other environmental factors, where what can occur in these situations is a greater allocation of biomass to the roots in order to maintain a good shoot/root ratio [50].

The similar results obtained for PE indicate that the cultivars showed similarity in the absorption, assimilation, and redistribution of N, and these processes are mediated by morphological, physiological factors, and nutritional demand during the development stages [51,52].

### 3.3. A Higher Agrophysiological Efficiency (APE) Index Indicates a Better Use of N

The results obtained for Altis 99 in APE and PE indicate that this cultivar, in addition to allocating N favoring biomass production, also permeates the N redistribution for good achene productivity. In light of this aspect, it is important to consider that the greatest accumulation of N in sunflowers occurs in the pre-flowering period [52]. However, to promote seed filling, aiming at greater productivity, the N remobilization of the vegetative parts occurs [2]. This remobilization of N to the achenes depends on a good supply of N to the plants [53]. Thus, Altis 99 presented good APE at lower N doses, indicating that this cultivar presents good efficiency in situations of low N availability, which can be attributed to its high productive potential [54].

Cultivar Aguará 06 had a higher APE, which may have occurred due to the partition of nitrogen compounds, demonstrating that the N applied in this cultivar is mainly directed to the production of achenes. Nitrogen compounds are essential for fruiting and are temporarily deposited in the vegetative part [55]. The use and partition of N are related to the efficiency of absorption and assimilation, as well as to a genetic factor, which permeates differentiations in relation to shoot and root biomass and the accumulation of N in shoots and achenes [56,57,58]. The results obtained for Aguará 06 in the APE demonstrate that this cultivar makes the best use of the N contained in the biomass and therefore presents high efficiency.

The highest APE obtained in the second agricultural harvest are related to the PE of that same harvest, since as the cultivars showed lower biological productivity, that is, less investment in vegetative growth, the production of achenes was favored. N is the main nutrient involved in increasing plant biomass, which promotes the accumulation of N in shoots [59]. However, at the fruiting development stage, this N is redistributed to ensure good productivity [55], which possibly occurred in the present work.

### 3.4. The Apparent Efficiency of Recovery (ARE) Is Influenced by the N Source and Is Reduced with Increasing Doses

The apparent recovery efficiency (ARE) is generally higher at lower doses of N [60], as observed in the present study. Cultivars BRS 122 and Altis 99 had higher ARE in lower doses. However, the highest dose provided a small increase in the ARE, indicating that at high doses of N, vegetative growth promotes the accumulation of N in the shoot and increases the apparent efficiency of recovery. The highest ARE obtained at lower doses directly influences the productivity of achenes, and the selection of cultivars from the ARE aims to increase UE and species productivity [61,62]. This trend of increasing ARE at lower doses has been observed in other studies [61,63].

For Multissol and Aguará 06, the ARE was higher at the lowest doses. As the N was increased, it reduced the ARE of these cultivars. The increase in ARE tends to increase due to the increase in biomass due to more efficient absorption of N. At lower doses, this can occur through stronger root growth, which enhances absorption and allows for greater N capture [64].

Altis 99 and Aguará 06 for this efficiency obtained higher values with the increase in N, indicating that these cultivars accumulate more nitrogen in the biomass when fertilized with larger amounts of urea; this accumulation of N is directly related to the amount of dry matter found [65]. In addition, the N source used influences the ARE, where ammonium sulfate and urea show better results for this efficiency compared to manure [66].

N recovery was more efficient in Multissol and BRS 122 in 2017, following the trend of higher ARE at lower N doses. The results obtained indicate that for these sunflower cultivars, N is more recovered at lower doses. The accumulation of N in the aerial part observed from the ARE is desirable, because during the active growth phase, the N is absorbed and accumulated in the vegetative parts; however, as the plant demand increases, this N is transferred to the achenes, and the yield therefore increases [57].

### 3.5. Utilization Efficiency (UE) Correlates to PE and ARE

Utilization efficiency (UE) relates to PE and ARE, involving total biological productivity, N accumulation in aboveground biomass, and sunflower achene productivity. UE was reduced at higher doses, following the trend of PE and ARE. The reduction in UE at higher doses of N indicates that these sunflower cultivars do not efficiently use N at higher doses; that is, higher doses exceed the acceptance rate of sunflower [57,58]. Some studies have found the same trend of reduced N absorption at higher doses [58].

The UE of Multissol and BRS 122 increased at the dose of 30 and 90 kg of N ha^−1^; the observed variation can be explained by a wide genetic variability influencing the UE, including characteristics such as total N uptake, N translocation, and N assimilation [67,68,69]. As sunflower responses to N availability vary according to the cultivar and the level of fertilizer used; however, UE follows the trend of ARE, being more responsive under conditions of low N availability [69,70].

The UE of cultivars decreased in 2017, indicating that in this season, the conditions imposed on the plants reduced the efficiency of N. Among the factors that limit the use of nitrogen fertilizers, high application rates are one of the main ones causing economic losses [71]. Therefore, the results obtained from the UE are useful to enhance the efficiency of the use of nitrogen fertilizers, as well as to reduce losses of these fertilizers that can cause environmental degradation [64].

## 4. Materials and Methods

### 4.1. Location and Characterization of the Experimental Area

The experiments were conducted in the field, at the Experimental Farm Rafael Fernandes (5°03′37″ S, 37°23′50″ W Gr and with an approximate altitude of 72 m), belonging to the Universidade Federal Rural do Semi-Árido (UFERSA), Mossoró-RN, from February to May 2016 and 2017. The region’s climate, according to the Köppen climate classification, is of the BSh type, with an average temperature of 27.2 °C and average annual rainfall of 766 mm [72]. The average meteorological data for the period of the experiments are shown in Figure 6.

The type of soil in the experimental area is classified as Abrupt Eutrophic Red-Yellow Latosol, texture-free sand [73]. Soil samples were collected at a depth of 0–20 cm for physical and chemical analysis, with the following physical characteristics: coarse sand = 660 g kg^−1^; fine sand = 220 g kg^−1^; silt = 20 g kg^−1^; clay = 100 g kg^−1^. The chemical characteristics of the soil after liming at a depth of 0–20 cm, 2016, and 2017 agricultural crops, respectively, were pH = 5.90 and 5.80; organic matter = 7.52 and 4.38 g kg^−1^; N = 0.42 and 0.32 g kg^−1^; P = 2.21 and 1.90 mg dm^−3^; K^+^ = 21.10 and 32.40 mg dm^−3^; Ca^2+^ = 0.40 and 1.40 cmol_c_ dm^−3^; Mg^2+^ = 0.57 and 0.70 cmol_c_ dm^−3^; Al^3+^ = 0.00 and 0.00 cmol_c_ dm^−3^.

### 4.2. Experimental Desing

The experimental design used in each experiment was completely randomized blocks, with four replications. The treatments were arranged in split plots, with five nitrogen rates (0, 30, 60, 90, and 120 kg ha^−1^ N) being allocated in the plots and the four sunflower cultivars (Aguará 06, Altis 99, Multissol, and BRS 122).

### 4.3. Conducting the Experiment

Soil preparation consisted of plowing, harrowing, and liming, which after 45 days fertilized the foundation, based on soil analysis and in accordance with the recommendations for the use of correctives and fertilizers [74]. The N source used was urea, supplied via irrigation water, through a diversion tank (“lung”). Fertigations with N were applied 1/3 at sowing and 2/3 split in the reproductive phase R1, with the appearance of a small floral bud at the apex and R3, when the elongation of the floral bud was at a distance greater than 2.0 cm above the last leaf, the splitting of N doses corresponded to the treatments, respectively.

Phosphorus was applied in the form of simple superphosphate, 70 kg ha^−1^ of P_2_O_5_, in the seeding hole, in the foundation fertilization. Potassium in the form of KCl, 70 kg ha^−1^ of K_2_O, was applied via fertigation according to nitrogen applications.

The total area of the experiment was 1008 m^2^, with each experimental plot consisting of four rows of plants with 0.30 m between plants and 0.70 m between rows, totaling an area of 12.6 m^2^ (4.5 × 2.8 m), in which the two central lines were considered, disregarding the plants at the ends, totaling a population of 47,619 plants ha^−1^.

The irrigation system was located by drip, with a spacing of 0.3 m between emitters and an average flow of 1.5 L h^−1^, using daily irrigation depth, considering rainfall and crop evapotranspiration (mm) in its phenological phases [75]. Rainfall during the sunflower phenological cycle accumulated 73 mm in 2016 and 188.4 mm in 2017, respectively.

The sowing of sunflowers in the first agricultural year was carried out on 23 February 2016, while in the second, it was carried out on 24 February 2017, performed manually at a depth of 4 cm, using 3 seeds per hole. The thinning was carried out 10 days after sowing, leaving one plant per hole. Weed management and phytosanitary control were carried out in accordance with technical recommendations and crop needs. 

In 2016, harvesting was carried out at phenological stage R9, which corresponded to 88 days after sowing (DAS) for cultivars BRS 122 and Multissol and 95 DAS for cultivars Altis 99 and Aguará 06. In 2017, for BRS 122 and Multissol at 90 DAS, and Altis 99 and Aguará 06 at 98 DAS, the bracts, capitulum, and stem showed dark brown color and achene moisture content of approximately 14 and 18%.

### 4.4. Variables Analyzed

The heads of all plants in the experimental area were collected and then dried. After drying, the achenes were threshed and cleaned. The harvesting, trailing, and cleaning of the achenes were carried out manually. The productivity of achenes was calculated by the mass of achenes in an experimental plot, which was corrected to 13% moisture and transformed into kg ha^−1^.

At the time of harvest, 4 plants from the useful area were collected and divided into stem, leaf, and capitulum; then, the washing process was carried out, and, subsequently, the plant material was dried in an oven at 65°C for approximately 48 h or until a constant mass was obtained. After drying, they were weighed to obtain the dry mass in grams. The total dry mass of the plant was considered the sum of the dry mass of the leaf, stem, and capitulum. Afterward, the results were converted to g ha^−1^, multiplying the result by the plant population and then to kg ha^−1^.

The dry mass of each vegetable component was ground in a Wiley electric mill, equipped with a stainless-steel sieve, until the material became homogeneous. Then, the material was packed in plastic bags for subsequent chemical analysis of the nutrient content. Subsequently, sulfuric digestion was carried out to determine the N, in which the total N was determined by titration of the distillate with a standardized solution of H2_S_O_4_ [76]. To determine the amount accumulated in each fraction of the plant, the concentration was multiplied by the dry mass of that fraction. To estimate the amount of total nutrients accumulated by the crop (kg ha^−1^) at the end of the cycle, the concentration of the accumulated nutrient in the plant was multiplied by the population density.

The nutrient use efficiency (UNE) by sunflower crop was estimated through agronomic efficiency (AE), physiological efficiency (FE), agrophysiological efficiency (AFE), apparent recovery efficiency (ARE), and utilization efficiency (UE), adapted from the methodology of [77,78,79].

The agronomic efficiency (AE) in the use of applied N was estimated by the relationship between the yield of achenes with and without N application and the amount of N applied, in kg kg^−1^ (Equation (1)):AE = (PAcP − PAsP)/QPa(1)
where PA_cN_ (kg) is the yield of sunflower achenes with N application; PA_sN_ (kg) is the yield of sunflower achenes without N application; and QN_a_ is the amount of N applied (kg).

The physiological efficiency (PE) of sunflowers was estimated by the relationship between shoot biomass with and without N application and N accumulation in shoot biomass with and without N application, in kg kg^−1^ (Equation (2)):PE = (PBcP − PBsP)/(APBcP − APBsP)(2)
where PB_cN_ (kg) is the total biological productivity (stem, leaves, and capitulum) with N application; PB_sN_ (kg) is the total biological productivity (stem, leaves and capitulum) without N application; ANB_cN_ (kg) is the accumulation of N in the aboveground biomass (stem, leaves and capitulum) with the application of N; and ANB_sN_ (kg) is the accumulation of N in the aboveground biomass (stem, leaves, and capitulum) without the application of N.

Agrophysiological efficiency (APE) was estimated by the relationship between sunflower achene yield with and without N application and N accumulation in shoot biomass with and without N application, in kg kg^−1^ (Equation (3)):
APE = (PAcP − PAsP)/(APBcP − APBsP)(3)
where PA_cN_ is the yield of sunflower achenes with N application; PA_sN_ is the yield of sunflower achenes without N application; ANB_cN_ is the accumulation of N in the aboveground biomass (stem, leaves, and capitulum) with the application of N, and ANB_sN_ is the accumulation of N in the aboveground biomass (stem, leaves, and capitulum) without the application of N.

The apparent recovery efficiency (ARE) was estimated by the relationship between the accumulation of N in the aboveground biomass with and without N application and the amount of N applied, in % (Equation (4)):ARE = (APBcP − APBsP/QPa) × 100(4)
where APB_cN_ is the accumulation of N in the aboveground biomass (stem, leaves, and capitulum) with the application of N and ANB_sP_ is the accumulation of N in the aboveground biomass (stem, leaves, and capitulum) without the application of N; and QN_a_ is the amount of N applied.

The utilization efficiency (UE) is the relationship between the physiological efficiency (pE) and the apparent recovery efficiency (ARE), in kg kg^−1^ (Equation (5)):UE = PE × ARE(5)

### 4.5. Statistical Analysis

Analyses of variance of agricultural crops were carried out separately for all characteristics evaluated using the SISVAR 5.6 application [80]. After observing the homogeneity of variances between agricultural crops, a joint analysis of these same characteristics was applied [81].

The response curve fitting procedure was performed using the Table Curve 2D program [82], with graphs created in SigmaPlot 12.0 [83]. Analysis of variance was performed to verify significance between treatments, and then the Tukey’s test (*p* < 0.05) was used to compare the means for sunflower cultivars and each agricultural season.

## 5. Conclusions

The dependence of modern agriculture on the use of chemical fertilizers requires attention to the efficient use of nutrients in decision-making.

The cultivar Altis 99 at a dose of 30 kg of N ha^−1^ showed the best results for agrophysiological efficiency and efficiency of use in the first crop.

In the second crop, the multisol cultivar at a dose of 30 kg ha^−1^ presented superior results for agronomic efficiency, agrophysiological efficiency, and utilization efficiency.

It is recommended that cultivars Altis 99 and Multissol are cultivated at a dose of 30 kg of N ha^−1^ in semiarid regions.

For subsequent studies, we indicate that there is a need to deepen the knowledge of the morphophysiological mechanisms that promote the distinct responses between sunflower cultivars.

## Figures and Tables

**Figure 1 plants-11-02390-f001:**
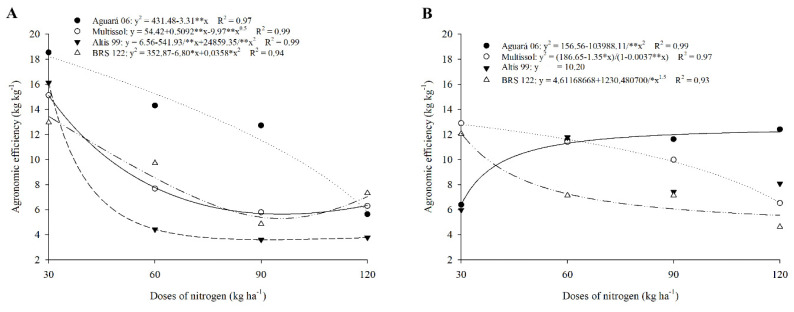
Agronomic efficiency of sunflower cultivars as a function of nitrogen doses in season 1 (**A**) and season 2 (**B**).

**Figure 2 plants-11-02390-f002:**
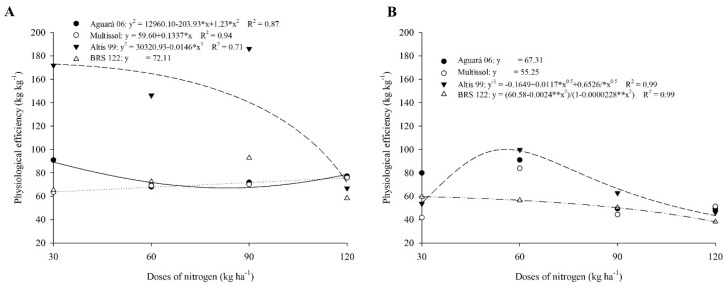
Physiological efficiency of sunflower cultivars as a function of nitrogen doses in season 1 (**A**) and season 2 (**B**).

**Figure 3 plants-11-02390-f003:**
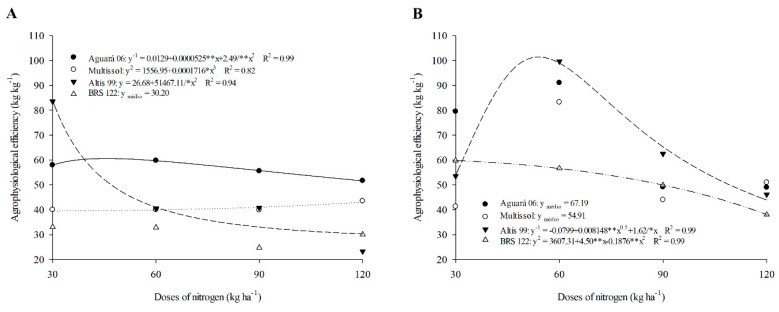
Physiological efficiency of sunflower cultivars as a function of nitrogen doses in season 1 (**A**) and season 2 (**B**).

**Figure 4 plants-11-02390-f004:**
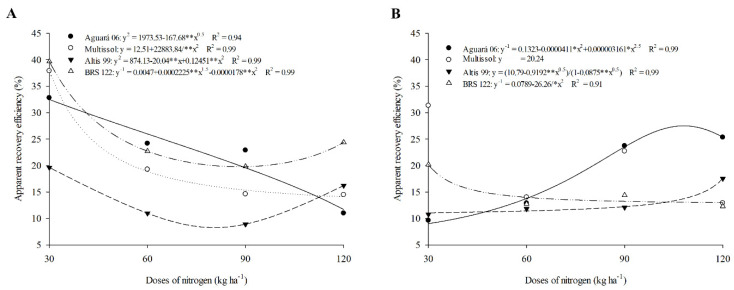
Apparent recovery efficiency of sunflower cultivars as a function of nitrogen doses in season 1 (**A**) and season 2 (**B**).

**Figure 5 plants-11-02390-f005:**
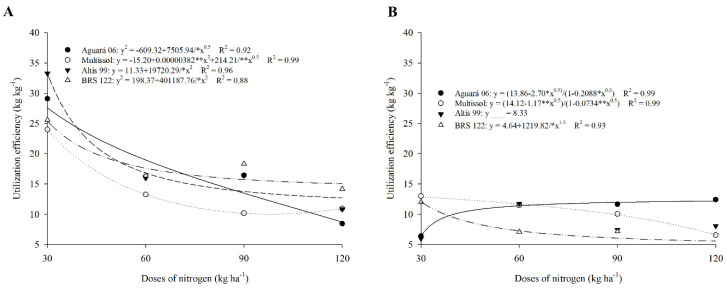
Utilization efficiency of sunflower cultivars as a function of nitrogen doses in season 1 (**A**) and season 2 (**B**).

**Figure 6 plants-11-02390-f006:**
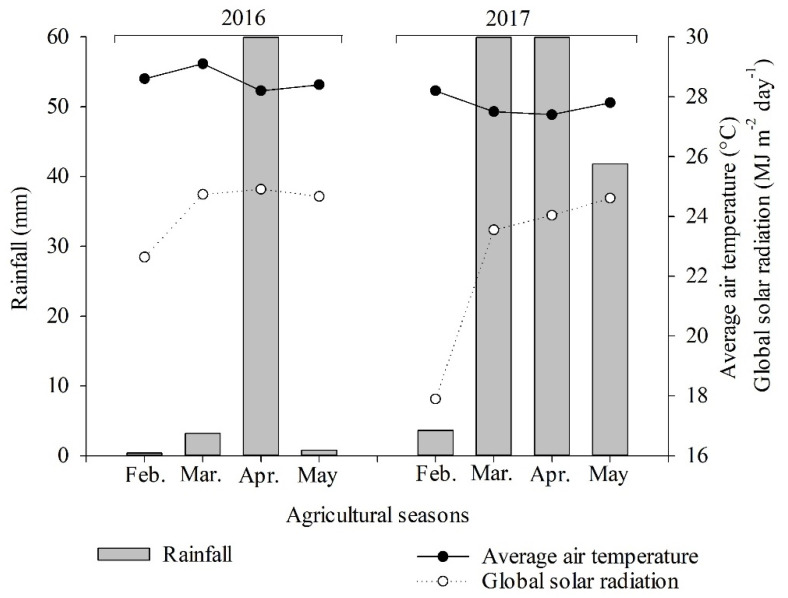
Average daily values of rainfall (mm), average air temperature (°C), relative air humidity (%), global solar radiation (MJ m^−2^ day^−1^), and phenological phases (emergency-VE, reproductive stage-R1 and harvest-R9) corresponding to the months of February to May of the 2016 and 2017 agricultural harvests.

**Table 1 plants-11-02390-t001:** Average values for agronomic efficiency as a function of sunflower cultivars within nitrogen doses in the 2016 and 2017 agricultural season.

**Agronomic Efficiency in the 2016 Agricultural Harvest**				
**Grow Crops**	**Nitrogen Dose (kg ha^−1^)**			
	**30**	**60**	**90**	**120**
Aguará 06	18.53 aA	14.30 aA	12.72 aA	5.64 abB
Multissol	15.12 bcA	7.68 bB	5.79 bB	6.29 abA
Altis 99	16.12 abA	4.43 cB	3.60 bB	3.77 bB
BRS 122	12.96 cA	9.71 bA	4.86 bB	7.32 aA
**Agronomic Efficiency in the 2017 Agricultural Harvest**				
**Grow crops**	**Nitrogen Dose (kg ha^−1^)**			
	**30**	**60**	**90**	**120**
Aguará 06	6.39 bB	11.52 aB	11.62 aA	12.40 aA
Multissol	12.89 aB	11.40 aA	9.98 aA	6.53 bcA
Altis 99	5.98 bB	11.77 aA	7.41 bA	8.07 bA
BRS 122	12.04 aA	7.14 bB	7.14 bA	4.63 cB

Means followed by the same lowercase letter in the column compare cultivars in the same harvest and means followed by the same capital letter compare cultivars in agricultural harvest; they do not differ from each other, by Tukey’s test, at 5% probability.

**Table 2 plants-11-02390-t002:** Mean values for the physiological efficiency as a function of sunflower cultivars within nitrogen doses in agricultural crops.

**Physiological Efficiency in the 2016 Agricultural Harvest**				
**Grow Crops**	**Nitrogen Dose (kg ha^−1^)**			
	**30**	**60**	**90**	**120**
Aguará 06	90.91 bA	67.95 bB	71.78 bA	77.18 aA
Multissol	63.49 cA	68.96 bA	70.05 bA	75.73 aA
Altis 99	171.85 aA	146.25 aA	186.17 aA	66.86 aA
BRS 122	65.07 cA	72.39 bA	92.72 bA	58.20 aA
**Physiological Efficiency in the 2017 Agricultural Harvest**				
**Grow Crops**	**Nitrogen Dose (kg ha^−1^)**			
	**30**	**60**	**90**	**120**
Aguará 06	79.89 aA	91.00 aA	49.27 aB	49.07 aB
Multissol	41.66 bB	83.77 aA	44.31 aB	51.24 aB
Altis 99	53.66 bB	99.75 aB	62.76 aB	46.20 aB
BRS 122	59.58 abA	56.50 bA	50.10 aB	38.04 aB

Means followed by the same lowercase letter in the column compare cultivars in the same harvest, and means followed by the same capital letter compare cultivars in the agricultural harvest; they do not differ from each other, by Tukey’s test, at 5% probability.

**Table 3 plants-11-02390-t003:** Mean values for agrophysiological efficiency as a function of sunflower cultivars within nitrogen doses in agricultural crops.

**Agrophysiological Efficiency in the 2016 Agricultural Harvest**				
**Grow Crops**	**Nitrogen Dose (kg ha^−1^)**			
	**30**	**60**	**90**	**120**
Aguará 06	57.98 bB	59.79 Ab	55.60 aA	51.69 aA
Multissol	40.06 bcA	40.04 aB	39.93 abA	43.53 abA
Altis 99	83.56 aA	40.52 aB	40.77 abB	23.28 bB
BRS 122	33.10 aB	42.87 aB	24.77 bB	30.06 bA
**Agrophysiological Efficiency in the 2017 Agricultural Harvest**				
**Grow Crops**	**Nitrogen Dose (kg ha^−1^)**			
	**30**	**60**	**90**	**120**
Aguará 06	79.54 aA	91.05 aA	49.15 aA	49.00 aA
Multissol	41.33 bA	83.29 aA	44.01 aA	50.98 aA
Altis 99	53.53 bB	99.66 aA	62.51 aA	46.15 aA
BRS 122	59.75 abA	56.64 bA	49.87 aA	38.02 aA

Means followed by the same lowercase letter in the column compare cultivars in the same harvest, and means followed by the same capital letter compare cultivars in agricultural harvest; they do not differ from each other, by Tukey’s test, at 5% probability.

**Table 4 plants-11-02390-t004:** Mean values for apparent recovery efficiency as a function of sunflower cultivars within nitrogen doses in agricultural crops.

**Apparente Recovery Efficiency in the 2016 Agricultural Harvest**				
**Grow Crops**	**Nitrogen Dose (kg ha^−1^)**			
	**30**	**60**	**90**	**120**
Aguará 06	32.78 bA	24.17 aA	22.87 aA	10.98 cB
Multissol	37.89 aA	19.26 bA	14.60 bB	14.46 bcA
Altis 99	19.61 cA	10.97 cA	8.90 cB	16.21 bA
BRS 122	39.63 aA	22.68 abA	19.79 aA	24.38 aA
**Apparent Recovery Efficiency in the 2017 Agricultural Harvest**				
**Grow Crops**	**Nitrogen Dose (kg ha^−1^)**			
	**30**	**60**	**90**	**120**
Aguará 06	9.61 cB	12.88 aB	23.72 aA	25.32 aA
Multissol	31.33 aB	14.00 aB	22.71 aA	12.89 cA
Altis 99	10.78 cB	11.82 aA	12.06 bA	17.52 bA
BRS 122	20.17 bB	12.72 aB	14.37 bB	12.28 cB

Means followed by the same lowercase letter in the column compare cultivars in the same harvest, and means followed by the same capital letter compare cultivars in agricultural harvest; they do not differ from each other, by Tukey’s test, at 5% probability.

**Table 5 plants-11-02390-t005:** Mean values for utilization efficiency as a function of sunflower cultivars within nitrogen doses in agricultural crops.

**Utilization Efficiency in the 2016 Agricultural Harvest**				
**Grow Crops**	**Nitrogen Dose (kg ha^−1^)**			
	**30**	**60**	**90**	**120**
Aguará 06	29.08 aA	16.23 aA	16.42 aA	8.45 bB
Multissol	24.01 cA	13.26 bA	10.18 bA	10.95 bA
Altis 99	33.26 bA	15.99 aA	16.41 aA	10.83 bA
BRS 122	25.59 cA	16.39 aA	18.30 aA	14.18 aA
**Utilization Efficiency in the 2017 Agricultural Harvest**				
**Grow Crops**	**Nitrogen Dose (kg ha^−1^)**			
	**30**	**60**	**90**	**120**
Aguará 06	6.42 bB	11.51 aB	11.65 aB	12.42 aA
Multissol	13.00 aB	11.47 aA	10.05 aA	6.56 bcB
Altis 99	5.99 bB	11.78 aB	7.44 bB	8.08 bB
BRS 122	12.01 aB	7.12 bB	7.17 bB	4.63 cB

Means followed by the same lowercase letter in the column compare cultivars in the same harvest, and means followed by the same capital letter compare cultivars in agricultural harvest; they do not differ from each other, by Tukey’s test, at 5% probability.

## Data Availability

Not applicable.

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
