# Peer review of "Efficiency of Nitrogen Use in Sunflower"

_plants, 2022, doi:10.3390/plants11182390_

Round 1

Reviewer 1 Report

I have reviewed with interest your manuscript entitled „Efficiency of nitrogen use in sunflower” submitted to Plants.

Authors in their study evaluated the analyze the agronomic yield of sunflower cultivars as a function of nitrogen fertilization; indicate the cultivar that has the best use of nitrogen. Moreover the main aim of the research was the measure the adequate N dose for sunflower through nutritional efficiency.

In my opinion the current version of your manuscript is suitable for publication in Plants, but after same revisions. The quality of the presentation could be improved-e.g. In general, manuscript needs small improvement.

The article suffers from a number of small mistakes, ranging from misspellings to incorrectly phrased sentences.

Some adjustments are suggested to qualify the paper:

Issues include:

The Abstract

The abstract should be a total of about 200 words maximum. Now it exceed 200 words (260 words).                        I propose to shorten it.

The general comment to the Introduction section: the introduction is written in an appropriate manner. The content of the literature review chapter is related to the research topic. Up-to-date literature references are presented in the manuscript by the author, but there are same references before 2010 – 30%.

In the chapter Materials and Methods, the methodology is adequate, but there is a lack of information in some aspects. Some more information should be explained in the text. It could be add the information with chemical content of soil before sowing sunflower. Please provide references on the methods used to provide the soil properties.

In the chapter Results, the results are displayed correctly.

The Discussion is informative. Moreover, the Authors attempt to discuss their important results and the rest is a quotation of literature. I suggest a small reorganization of Discussion, and include more up-to-date literature.

In my opinion the Conclusions is insufficiency. It could be contain more obtained by Authors results and same recommendations for farmers and other recipient of study results.

More small recommendation there are in the text of manuscript.

I hope that these comments help you to make an improved the final version of the manuscript.

Author Response

REVISION NOTES

Dear Editors,

In first place, thank you for the critical reviewers’ comments that helped me to improve the present revision. Below are listed all the modifications required. The manuscript was strong revised according to your suggestions:

Reviewer 1

ABOUT THE REVIEWER'S COMMENT:

Abstract: The authors can improve the wording in their manuscript

Line 16: I suggest that the authors replace “ideal” with “optima”

Line 18: If I were you, I would use “…cultivars with a high nitrogen use efficiency…” instead of the best…

Line 20: Use a direct expression such as “The split plot design was used to conduct the experiments”…

Line 23: Consider revising this statement. For instance, “To estimate the NUE of cultivars, we measured the AE, PE, etc.

Line 24: I suggest to start the phrase with “results or data indicate that…”

REPLY: We accept and correct the suggestions made.

 Line 27: is the dosage of 30 Kg of N ha-1 higher or lower than commonly applied N for sunflower? A statement in this sense is necessary to give some insight.

REPLY: The dosage of N varies according to the region of cultivation. However, most studies with sunflower indicate a linear increase in productivity with increasing doses.

Line 28: what does EU stands for?

REPLY: Utilization efficiency.

 Line 29–30: This conclusive statement can be improved to a more suggestive one and put in a general perspective of nitrogen application rates in sunflower.

REPLY: We accept and correct the suggestions made.

INTRODUCTION

Lines 35–36: Consider improving this introductive statement. It does not sound clear. Line 44: I recommend the authors to rephrase this statement and provide a more meaningful one with specific message. Correct “incPreased" to increased

Line 58: write properly the chemical form of ammonium (NH4+) and nitrate (NO3–)

Line 64: remove the double space between …efficiency and with…

Line 73: could you use a different word with a similar meaning “disturbances”

Lines 63, 75, etc. I suggest that the authors use optimum instead of ideal

Line 89: …use “could be due”…instead of “is due”…

Line 92: when citing a reference at the beginning of a sentence, show the authors name in reference [40] (i.e. Juan et al. [40] studied…, and found that…). Apply the same change in line 94 (reference [41], etc.).

Lines 95–99: The statement of the hypothesis should be improved. The authors should consider indicating the aim of the study, and then tell what was done to verify their hypotheses.

REPLY: We accept and correct the suggestions made.

Reviewer 2 Report

The conclusions are too weak compared to the article. For example, the manuscript analyzes 4 cultivars of sunflower, but there are recommendations  only for two. The conclusions should reflect the nutrient use efficiency (UNE) in general, not only separate indices such AE, PE, etc. 

Author Response

We accept and correct the suggestions made.

Reviewer 3 Report

Coêlho et al. have challenged to investigate the nitrogen use efficiency of Sunflower cultivars using different N doses. This study, placed in the context of climate change, provides some useful information and could be exploited to optimize N use in Sunflower and reduce excessive application of  N  to lower environmental impact. However, this study contains some issues that require a close attention and be clarified to provide more insights.

Here are my comments

Abstract: The authors can improve the wording in their manuscript

Line 16: I suggest that the authors replace “ideal” with “optima”

Line 18: If I were you, I would use “…cultivars with a high nitrogen use efficiency…” instead of the best…

Line 20: Use a direct expression such as “The split plot design was used to conduct the experiments”…

Line 23: Consider revising this statement. For instance, “To estimate the NUE of cultivars, we measured the AE, PE, etc.

Line 24: I suggest to start the phrase with “results or data indicate that…”

Line 27: is the dosage of 30 Kg of N ha-1 higher or lower than commonly applied N for sunflower? A statement in this sense is necessary to give some insight.

Line 28: what does EU stands for?

Line 2930: This conclusive statement can be improved to a more suggestive one and put in a general perspective of nitrogen application rates in sunflower.

Introduction

Lines 3536: Consider improving this introductive statement. It does not sound clear.

Line 44: I recommend the authors to rephrase this statement and provide a more meaningful one with specific message. Correct “incPreased" to increased

Line 58: write properly the chemical form of ammonium (NH4+) and nitrate (NO3)

Line 64: remove the double space between …efficiency and with…

Line 73: could you use a different word with a similar meaning “disturbances”

Lines 63, 75, etc. I suggest that the authors use optimum instead of ideal

Line 89: …use “could be due”…instead of “is due”…

Line 92: when citing a reference at the beginning of a sentence, show the authors name in reference [40] (i.e. Juan et al. [40] studied…, and found that…). Apply the same change in line 94 (reference [41], etc.).

Lines 9599: The statement of the hypothesis should be improved. The authors should consider indicating the aim of the study, and then tell what was done to verify their hypotheses.

Results

-          I found the description of the results minimalistic. The authors should give more insight into their find their findings and critically analyze their results. And prior to describe the results, a brief context should be provided to help the readers understand why these traits are important.

-          In addition, all figures captions should be improved, while providing the meaning of the information in different panels.

-          I am also curious to know the lettering in tables represent what, and why the difference in cases (capital and small). Did the authors analyzed the data according the split plots experimental design initially applied to perform the experiments?

-          I recommend the authors to check the correlation between traits to see whether they influence each other or not.

-          Did the authors measured the yield of the tested Sunflower cultivars under different N application?

-          I did not see the soil properties and nutrients availability prior to applying different N levels.

Discussion

All subtitles in the discussion section should deliver a clear message emphasizing on the critical findings. Reproducing the title from the results do not help capture the message.

After improving the results section, I recommend the authors to critically discuss their findings in a broad context relative to other studies, while putting them into perspective. Consider improving.

Materials and Methods

Line 354: Figura 1 (?). This figure should be moved to supplementary material and be cited accordingly.

Lines 368372: There a huge contradiction that will confuse the readers. I am curious to really know which experimental design was used in to perform the experiments and the statistical analysis of the data. In line 20 (abstract), the authors indicated that the split plots design was use to perform the experiments. Now, in the M&M, it is the completely randomized block design but arranged in a split plots. I would like to remind that CRBD and Split plots are two distinct experimental designs and should treated as such. What did the authors really used?

Conclusion

The conclusion is weak and should be improved. It should give a take-home message for the readers, while highlighting the findings of this study.

Efficiency of nitrogen use in sunflower

Manuscript ID: plants-1829363

Coêlho et al. have challenged to investigate the nitrogen use efficiency of Sunflower cultivars using different N doses. This study, placed in the context of climate change, provides some useful information and could be exploited to optimize N use in Sunflower and reduced exogenous applications of N excessively to reduce environmental impact. However, this study contains some issues that require a close attention and be clarified to provide more insights.

Here are my comments

Abstract: The authors can improve the wording in their manuscript

Line 16: I suggest that the authors replace “ideal” with “optima”

Line 18: If I were you, I would use “…cultivars with a high nitrogen use efficiency…” instead of the best…

Line 20: Use a direct expression such as “The split plot design was used to conduct the experiments”…

Line 23: Consider revising this statement. For instance, “To estimate the NUE of cultivars, we measured the AE, PE, etc.

Line 24: I suggest to start the phrase with “results or data indicate that…”

Line 27: is the dosage of 30 Kg of N ha-1 higher or lower than commonly applied N for sunflower? A statement in this sense is necessary to give some insight.

Line 28: what does EU stands for?

Line 2930: This conclusive statement can be improved to a more suggestive one and put in a general perspective of nitrogen application rates in sunflower.

Introduction

Lines 3536: Consider improving this introductive statement. It does not sound clear.

Line 44: I recommend the authors to rephrase this statement and provide a more meaningful one with specific message. Correct “incPreased" to increased

Line 58: write properly the chemical form of ammonium (NH4+) and nitrate (NO3)

Line 64: remove the double space between …efficiency and with…

Line 73: could you use a different word with a similar meaning “disturbances”

Lines 63, 75, etc. I suggest that the authors use optimum instead of ideal

Line 89: …use “could be due”…instead of “is due”…

Line 92: when citing a reference at the beginning of a sentence, show the authors name in reference [40] (i.e. Juan et al. [40] studied…, and found that…). Apply the same change in line 94 (reference [41], etc.).

Lines 9599: The statement of the hypothesis should be improved. The authors should consider indicating the aim of the study, and then tell what was done to verify their hypotheses.

Results

-          I found the description of the results minimalistic. The authors should give more insight into their find their findings and critically analyze their results. And prior to describe the results, a brief context should be provided to help the readers understand why these traits are important.

-          In addition, all figures captions should be improved, while providing the meaning of the information in different panels.

-          I am also curious to know the lettering in tables represent what, and why the difference in cases (capital and small). Did the authors analyzed the data according the split plots experimental design initially applied to perform the experiments?

-          I recommend the authors to check the correlation between traits to see whether they influence each other or not.

-          Did the authors measured the yield of the tested Sunflower cultivars under different N application?

-          I did not see the soil properties and nutrients availability prior to applying different N levels.

Discussion

All subtitles in the discussion section should deliver a clear message emphasizing on the critical findings. Reproducing the title from the results do not help capture the message.

After improving the results section, I recommend the authors to critically discuss their findings in a broad context relative to other studies, while putting them into perspective. Consider improving.

Materials and Methods

Line 354: Figura 1 (?). This figure should be moved to supplementary material and be cited accordingly.

Lines 368372: There a huge contradiction that will confuse the readers. I am curious to really know which experimental design was used in to perform the experiments and the statistical analysis of the data. In line 20 (abstract), the authors indicated that the split plots design was use to perform the experiments. Now, in the M&M, it is the completely randomized block design but arranged in a split plots. I would like to remind that CRBD and Split plots are two distinct experimental designs and should treated as such. What did the authors really used?

Conclusion

The conclusion is weak and should be improved. It should give a take-home message for the readers, while highlighting the findings of this study.

Author Response

REVISION NOTES

Dear Editors,

In first place, thank you for the critical reviewers’ comments that helped me to improve the present revision. Below are listed all the modifications required. The manuscript was strong revised according to your suggestions:

Reviewer 3

RESULTS AND DISCUSSION

ABOUT THE REVIEWER'S COMMENT:

I found the description of the results minimalistic. The authors should give more insight into their find their findings and critically analyze their results. And prior to describe the results, a brief context should be provided to help the readers understand why these traits are important.

REPLY: The authors considered that the results and discussions are adequate, we explain all the data in the two seasons and emphasize the indices for each cultivar.

- In addition, all figures captions should be improved, while providing the meaning of the information in different panels.

REPLY: We accept and correct the suggestions made.

- I am also curious to know the lettering in tables represent what, and why the difference in cases (capital and small). Did the authors analyzed the data according the split plots experimental design initially applied to perform the experiments?

- I recommend the authors to check the correlation between traits to see whether they influence each other or not.

REPLY: We accept and correct the suggestions made.

- Did the authors measured the yield of the tested Sunflower cultivars under different N application?

REPLY: The yield used in the study was based on dry matter.

- I did not see the soil properties and nutrients availability prior to applying different N levels

REPLY: This information is in the methodology: Soil samples were collected at a depth of 0 - 20 cm for physical and chemical analysis. With the following physical characteristics: coarse sand = 660 g kg-1; fine sand = 220 g kg-1; silt = 20 g kg-1; clay = 100 g kg-1. The chemical characteristics of the soil after liming at a depth of 0 - 20 cm, 2016 and 2017 agricultural crops, respectively, were: pH = 5.90 and 5.80; organic matter = 7.52 and 4.38 g kg-1; N = 0.42 and 0.32 g kg-1; P = 2.21 and 1.90 mg dm-3; K+ = 21.10 and 32.40 mg dm-3; Ca2+ = 0.40 and 1.40 cmolc dm-3; Mg2+ = 0.57 and 0.70 cmolc dm-3; Al3+ = 0.00 and 0.00 cmolc dm-3.

Materials and Methods

-Line 354: Figura 1 (?). This figure should be moved to supplementary material and be cited accordingly.

REPLY: Figure 1 should not be transferred to the complementary material, as it provides necessary information that demonstrates the influence of precipitation on the measured indices.

- Lines 368–372: There a huge contradiction that will confuse the readers. I am curious to really know which experimental design was used in to perform the experiments and the statistical analysis of the data. In line 20 (abstract), the authors indicated that the split plots design was use to perform the experiments. Now, in the M&M, it is the completely randomized block design but arranged in a split plots. I would like to remind that CRBD and Split plots are two distinct experimental designs and should treated as such. What did the authors really used?

REPLY: The experimental design used in each experiment was completely randomized blocks, with four replications. The treatments were arranged in split plots, with five nitrogen rates (0, 30, 60, 90, and 120 kg ha-1 N) being allocated in the plots and the four sunflower cultivars (Aguará 06, Altis 99, Multissol, and BRS 122).

CONCLUSION

The conclusion is weak and should be improved.

REPLY: We accept and correct the suggestions made.

Round 2

Reviewer 3 Report

The authors have attempted to address the concern raised in the previous version.  Below is the reply by the authors:

REPLY: The experimental design used in each experiment was completely randomized blocks, with four replications. The treatments were arranged in split plots, with five nitrogen rates (0, 30, 60, 90, and 120 kg ha-1 N) being allocated in the plots and the four sunflower cultivars (Aguará 06, Altis 99, Multissol, and BRS 122).

Comment:

I understood that the authors used a split-plot in RCDB. If yes, consider improving the description in the materials and methods section for more clarity. In addition, I recommend the authors indicate that the ANOVA for RCDB was used to evaluate the statistical significance between treatments and sub-treatments. I am therefore asking the authors to perform a multiple comparison analysis in case the ANOVA showed a significant difference, in order to identify nitrogen levels with high statistical significance. Furthermore, I recommend the authors add a layout of their experimental design as supplementary materials, showing blocks and subplots randomized.

Author Response

REVISION NOTES

Dear Editors,

In first place, thank you for the critical reviewers’ comments that helped me to improve the present revision. Below are listed all the modifications required. The manuscript was strong revised according to your suggestions:

Reviewer

ABOUT THE REVIEWER'S COMMENT:

  1. The layout with the experiment information was added to the complementary material.
  2. The explanation of the stat has been revised and improved.

Round 3

Reviewer 3 Report

The authors have addressed the concerns raised, and the manuscript has been improved.